# Serum Adipocytokines Profile in Children Born Small and Appropriate for Gestational Age—A Comparative Study

**DOI:** 10.3390/nu15040868

**Published:** 2023-02-08

**Authors:** Justyna Zamojska, Katarzyna Niewiadomska-Jarosik, Agnieszka Wosiak, Marta Gruca, Elżbieta Smolewska

**Affiliations:** 1Department of Pediatric Cardiology and Rheumatology, Medical University of Lodz, 90-008 Lodz, Poland; 2Institute of Information Technology, Lodz University of Technology, 90-008 Lodz, Poland

**Keywords:** adipocytokines, adiponectin, leptin, resistin, children, small for gestational age

## Abstract

Background: Adipose tissue is not only a storage place for fat, but also an endocrine organ, secreting bioactive molecules which influence body metabolism. Such molecules are known as adipocytokines. In the past years the coincidence between adipocytokines and fetal growth restriction disorders was found. The aim of the study was to estimate serum levels of adiponectin, leptin and resistin in children born small for gestational age, compared to children born at an appropriate size for gestational age. Methods: The study consisted of 35 children aged seven to nine years, born SGA (small for gestational age) on term and 25 healthy children (14 girls, 11 boys), born with proper birthweight (AGA—appropriate for gestational age)–control group. Results: Adiponectin and leptin levels were significantly higher in the SGA group compared to the AGA group (*p* = 0.023, *p* = 0.018 respectively). The resistin values were comparable in both groups of patients. There was a positive correlation between serum leptin concentration and current body weight in SGA group (r = 0.28; *p* = 0.108). In turn, adiponectin levels in this group of patients negatively correlated with actual body weight (r = −0.51; *p* = 0.002). The negative correlation between body mass index and plasma adiponectin levels was found only in children born SGA. SGA children had significantly higher values of diastolic blood pressure. There was negative correlation between serum adiponectin level and systolic blood pressure in SGA children. In the SGA group the phenomenon of catch-up growth was observed in 32 children. Conclusions: Children born SGA have abnormal adipose tissue biomarkers profiles.

## 1. Introduction

According to the NCD Risk Factor Collaboration’s 2021 report, up to 213 million children worldwide are overweight and 124 million are obese [1]. During the COVID-19 pandemic, the number of overweight and obese children increased [2]. As can be seen from the papers cited above, obesity and its complications are already occurring in the pediatric population. In order to prevent the negative health outcomes of obesity, it is important to identify children at increased risk of metabolic disorders early and to put preventive actions in this group of patients. The pathogenesis of obesity is multifactorial, linked to genetic, psychosocial, socio-economic and environmental factors [3]. Among other factors, the metabolic dysregulation taking place in adipose tissue contributes to the development of obesity [4]. A group at increased risk of obesity in adulthood are pediatric patients born as SGA [5].

As is known from literature, adipose tissue may play a major role in linking poor fetal growth to later development of metabolic diseases in adulthood [6,7,8].Over the past decade it has been proved that adipose tissue is a complex organ that not only stores fat, but is also an endocrine organ, secreting plenty of bioactive molecules which influence body metabolism [9]. Such molecules derived from adipocytes are known as adipocytokines. In recent years the connection between adipocytokines and fetal growth restriction disorders was reported [10]. Over the years, adipokines have been shown to play a key role in the regulation of many important processes in the human body, such as insulin sensitivity, glucose and lipid metabolism, as well as immune and inflammatory responses [11].

Low serum levels of anti-inflammatory adiponectin and high concentrations of pro-inflammatory molecules, such as IL-6, TNF-α, CRP, resistin, and leptin, may be related to low fetal weight and may worsen this condition [10,12]. Adiponectin plasma levels in obese people are lower than in non-obese subjects [13]. Leptin is an adipokine directly correlated with the pro-inflammatory phenotype associated with obesity. It has been revealed that leptin inhibits adipogenesis, promotes fat demarcation, encourages chronic inflammation, increases insulin sensitivity, and stimulates angiogenesis [14]. High leptin levels are directly linked to obesity and later to the development of metabolic disease consequences such as cardiovascular disease, insulin resistance, and type 2 diabetes [15]. In addition, through pro-inflammatory effects, elevated serum leptin levels correlate with insulin resistance, arterial stiffness, elevated blood pressure [16]. In reverse, serum adiponectin levels have been found to be inversely related to obesity, thought to be anti-inflammatory, and its elevation is associated with decreased arterial stiffness and lower blood pressure [17].

Among obese adults, leptin plays a significant role in the development of hypertension in this group of patients [18]. There are still an insufficient number of studies examining this relationship in the pediatric population.

Numerous research studies suggest that many adipocytokines, such as leptin, adiponectin, resistin, or visfatin can be detected in the umbilical blood. A correlation is observed between adipocytokines and birthweight as well as age [19,20,21,22,23]. So far, few studies have been published on adipocytokines in pre-pubertal children born small for gestational age, and they are contrary to the results of this study [24,25,26,27,28]. Children born SGA due to an abnormal adipokine profile have a higher risk of metabolic syndrome and resulting complications in adulthood [29].

The purpose of this study was to evaluate serum levels of adiponectin, leptin and resistin in children born small for gestational age, compared to children born with normal gestational age and birthweight. Adipokine levels change during puberty, with adiponectin levels decreasing in parallel with the progression of puberty in children, more markedly in boys than in girls, and linked to hormonal changes, particularly plasma androgen levels [30]. It has been tentatively shown that male and female sex steroids have opposing effects on leptin secretion and tissue sensitivity to leptin, with testosterone showing inhibitory effects and estrogen showing stimulatory effects [31]. To reduce confounding factors, the study was conducted on a group of prepubertal children.

## 2. Materials and Methods

### 2.1. Source of Data

The analysis included 35 children (18 girls, 17 boys) aged 7–9 years, randomly selected from the database of the Obstetrics and Gynecology Department. From this database, children born at term who were SGA (small for gestational age), i.e., birth weight below the 10th percentile according to gestational age, detected prenatally by fetal size measurements on obstetric ultrasonography, were recruited to the study. The control group included 25 healthy subjects (14 girls, 11 boys), born with normal birthweights (AGA—appropriate for gestational age), who were matched to the study group by age and sex.

### 2.2. Survey

The study was performed at the Pediatric Cardiology and Rheumatology Department of the Medical University of Lodz. All subjects were well at the time of the study; none had a chronic illness or drug history (Table 1). 

All of the children were non-obese, based on the standard deviation score—SDS-of body mass index ≤ 2 for the chronological age and gender and were in the pre-pubertal period defined as Tanner 1. 

All studied patients also underwent manual oscillometric measurements of arterial pressure with the use of proper cuff size (proper to the length and circumference of the child’s arm). Three independent measurements were taken after at least ten minutes of rest in a seated position. All children were assessed according to the Polish norms of blood pressure for children and adolescents—OLAF study [32].

The exclusion criteria in mothers were: multiple gestation, gestational hypertension/pre-eclampsia, pre-existing (before pregnancy) diabetes, gestational diabetes, intrahepatic cholestasis of pregnancy, maternal infection with syphilis or HIV, systemic diseases (e.g., hypertension, diabetes, rheumatological diseases, thyroid diseases, chronic kidney diseases, obesity), and taking certain drugs (e.g., antiarrhythmic, epileptic, anti-inflammatory drugs, insulin). Also, maternal smoking or alcohol use or illicit drug use during pregnancy were exclusion criteria for participation in the study. The exclusion criteria in children were: evidence for chromosomal or infectious etiology of SGA, hypothyroidism, systemic or acute disease (e.g., kidney or liver dysfunction, hypertension, gastrointestinal, nephrological, neurological, or cardiac diseases). We did not exclude children who had histories of taking antibiotics or other symptomatic drugs used for infections. Children with other pharmacological treatment were excluded. Informed consent was acquired from all children’s parents. 

Gestational age was calculated according to mothers’ last menstrual period. Body weight was measured in subjects wearing light clothes and bare-footed with a calibrated scale. Blood samples were collected during the morning period from children who had fasted for 12 h.

Adiponectin, leptin, and resistin concentration in serum were measured by an enzyme-linked immunosorbent assay (ELISA kit by R & D Systems, MN 55413,USA). Minimum detectable concentrations of the ELISA kit for leptin—7.8 pg/mL (kit DLP00), for adiponectin—0.079–0.891 ng/mL (kit DRR300), and for resistin—0.010–0.055 ng/mL (kit DRSN00) were used. 

### 2.3. Statistical Analyses

Parameters distribution was tested for normality using Shapiro-Wilk W test. The differences in characteristics between the AGA group and the SGA patients were assessed using the Mann-Whitney-U test. Correlations were evaluated using the Spearman’s correlation coefficient. The results were regarded statistically significant if *p*-value was <0.05. “Statistica” software version 10 (Statsoft Polska, Kraków, Poland) was used for all performed statistical analysis.

This study was approved by Medical Ethical Committee of the Health Sciences Faculty of Lodz University (No: RNN/150/09/KB).

## 3. Results

Medical records analysis confirmed statistically significant difference in birthweight (*p* < 0.001), while there was no significant difference for gestational age between groups. 

According to oscillometric measurements of blood pressure, there was a statistically significant difference between the SGA and AGA groups in relation to diastolic blood pressure (63.72 ± 6.93 vs. 58.48 ± 9.71 mmHg, *p* < 0.05) but not in systolic blood pressure (102.64 ± 8.59 vs. 101.34 ± 9.96 mmHg, *p* > 0.05). All patients had the mean values of blood pressure in manual measurements below 95th percentile for sex and height therefore hypertension was not diagnosed in any of the examined children.

Adiponectin serum levels were significantly higher in SGA group compared to AGA group (*p* = 0.023). Leptin concentration was also significantly higher in SGA group than in healthy peers (*p* = 0.018). In turn, the resistin values were similar in both groups of patients (Table 2). 

There was an observed positive correlation between serum leptin concentration and current body weight in the SGA group, but it was not significant (r = 0.28; *p* = 0.108) (Figure 1). 

In relation to BMI, we also observed a positive correlation in both groups of patients, but also without statistical significance.

On the other hand, the serum level of adiponectin in the SGA group correlated negatively with actual body weight (r = −0.51; *p* = 0.002). In the AGA group this correlation was positive and not statistically significant. The statistically significant negative correlation between plasma adiponectin levels and body mass index (r = −0.42; *p* = 0.013) was observed only in children born SGA (Figure 2). In the AGA group this correlation was positive, but without statistical meaning (r = 0.134; *p* = 0.634).

Referring to measurements of blood pressure there was statistically significant negative correlation between systolic blood pressure and adiponectin serum levels in SGA group (r = −0.36; *p* = 0.03). The same correlation was found for diastolic blood pressure but it was not statistically significant (r = −0.11; *p* = 0.54). In AGA group there was also negative correlation observed between serum adiponectin levels and systolic and diastolic blood pressure, but without statistical significance. In both group of patients there was statistically not significant positive correlation between leptin serum levels and systolic and diastolic blood pressure. For resistin, the results were inconsistent. There was statistically not significant negative correlation between resistin serum levels and systolic (r = −0.32; *p* = 0.24) as well as diastolic (r = −0.42; *p* = 0.13) blood pressure measurements is AGA group. On the other hand in SGA group this correlation was negative for systolic blood pressure (r = −0.06; *p* = 0.72) and positive for diastolic blood pressure (r = 0.01; *p* = 0.97). None of these correlations were statistically significant.

Analyzing SGA children in relation to body weight gain and BMI (BMI-body mass index), the phenomenon of catch-up growth was observed in 32 children. Only three children were below average weight and BMI at the time of the examination.

## 4. Discussion

There are few reports in the literature on the levels of adipocytokines in people born SGA. Most of them relate to newborns, some teenagers, and some adults, and sometimes they describe conflicting results [19,20,26,27,28,33,34,35]. In addition, publications describing the concentration of adipocytokines in children present the results of research from over 10 years ago.

Our research studies on pre-pubertal age children are among the few in this age group. The present study indicates a significantly higher level of adipocytokines such as leptin and adiponectin in children aged seven to nine years, born SGA. In addition, we have shown that leptin has a positive correlation with current body weight (without statistical significance), while adiponectin correlates negatively with current body weight and body mass index. Our results are similar to those of Challa et al. who evaluated children aged 4 to 10 years [36]. The inverse relationship of adiponectin with BMI (body mass index) observed in the SGA group is comparable to our results. Leptin levels tend to have higher values in the SGA group as in our study, but there were no significant differences between groups. This may suggest that this hormone depends rather on the current body weight than on SGA status at birth.

On the other hand, Evagelidou et al. presented average leptin values that were higher in the SGA than in the AGA group; however, there were no statistically significant differences between groups [37]. 

In addition, Kistner et al. showed higher leptin levels in the SGA group; however, they were a slightly older age group—the examined children were 8.5 to 10 years old [38]. Finnish authors evaluated adiponectin levels in children aged mean 12.2 years. Their findings did not show differences between the SGA and AGA groups, whereas they showed a negative correlation of adiponectin with waist circumference [10]. Our study has presented a negative correlation between adiponectin and current body weight along with BMI in the SGA group. Cianfarani et al. describe different results [27]. They revealed lower levels of adiponectin in SGA children. However, this study concerned only short stature children with SGA and short stature as well as obese children with AGA. Additionally, in the Huang et al. study short SGA children had lower adiponectin levels than short AGA controls [28].

Regarding resistin concentrations, there are inconsistent findings. Some authors have confirmed higher resistin values in SGA infants [39], while other authors have observed lower values [40]. In our research, resistin levels did not show significant differences between the groups. Only a few reports refer to resistin levels in SGA children who were born on time according to gestational age, and they do not relate to children of pre-pubertal age. Among the others, Farid et al., evaluating resistin levels in the umbilical cord blood of 31 AGA newborns, 41 SGA neonates showed significantly higher levels of this adipocytokine in the SGA group [20]. Interesting results were shown by Deng et al. Their study included 56 children born SGA with catch-up growth and 55 born without catch-up growth, who were further divided into groups I (with BMI catch-up) and II (without BMI catch-up) and 52 children born appropriate for gestational age with normal height. The serum adiponectin levels of children born SGA without catch-up growth with BMI catch-up and SGA children without BMI catch-up with catch-up growth were significantly lower than those from the SGA group without BMI and catch-up growth. In this study lower levels of adiponectin were correlated with higher BMI and postnatal height catch-up growth in SGA children. However, researchers did not reveal significant differences when comparing various AGA groups. They concluded that lower levels of adiponectin were closely correlated with height and BMI catch-up growth in SGA children [41]. Our analysis showed a negative correlation of adiponectin with current body weight and body mass index. Results from a team of Korean researchers showed that adiponectin levels increased with age and pubertal development and were higher in prepubertal SGA children with weight catch-up than in children without weight catch-up [42]. Stawerska et al. evaluated the impact of ghrelin on achieving normal body height and the occurrence of metabolic complications in SGA children born at term. They observed in SGA children who achieved proper height significantly higher values of ghrelin and insulin growth factor-1 (IGF-1) and lower adiponectin values. It can be concluded that increased production of ghrelin and IGF-1 seems to be an adaptation mechanism in SGA children, enabling them to achieve normal growth in the future [43].

In the literature there are a number of studies evaluating adipocytokines levels in newborns and neonates. This level, evaluated from umbilical cord blood of newborns who were SGA, seems to have a different profile than later levels in the same group. Ziyang Zhu et al. investigated 38 neonates and showed significantly lower values of adiponectin in umbilical cord blood of SGA newborns [19]. Similar results were published in 2009 by Martos-Moreno et al. [44]. In the SGA newborns group, there were significantly lower adiponectin as well as leptin levels. Adiponectin correlated positively with gestational age, while leptin correlated positively with birth weight. According to these authors, gestational age and birth weight are the main determinants of adipocytokines cord blood profiles.

Mohamed et al. assessed umbilical serum adiponectin levels. In this study umbilical resistin levels were elevated in SGA newborns compared with control, and umbilical resistin and adiponectin were negatively correlated with birth weight, birth length, and head circumference [45]. 

Slightly different observations were made by Bozzola et al. [46]. In this study there were no differences in the levels of circulating adiponectin during examinations of SGA and AGA children in the neonatal period and at 1, 6 and 12 months of life, whereas circulating leptin levels were higher in AGA newborns, similar in one- and six-month-olds, and higher in the 12th month of life in the SGA group when children showed catch-up growth. Small for gestational age infants showed an increase in adipose tissue that is a typical phenomenon after a period of undernutrition. Therefore, authors speculate that the increased levels of leptin may derive from this increased adipose tissue. Leptin levels of all neonates at birth were positively correlated with the BMI. Furthermore, Buck et al. based on data obtained in the HOME study (Health Outcomes and Measures of the Environment Study), showed that in SGA neonates, cord leptin levels were lower than in AGA children. The concentration of adiponectin in cord blood was comparable in each group [47]. Visentin et al. analyzed umbilical serum leptin and adiponectin of intrauterine growth retardation in small for gestational age and appropriate for gestational age neonates. They found no differences in adiponectin and leptin concentrations between SGA and AGA groups [10].

The concentrations of leptin and adiponectin were positively correlated with birth weight [24].

It has been proven that the risk of cardiovascular morbidity and mortality over a lifetime is influenced by blood pressure values during childhood [48].

Infants born with SGA are reported to have a higher risk of developing high blood pressure, likely due to inadequate organ development in utero [49]. In our study all children had normal values of blood pressure according to polish norms (OLAF study), but we find differences in relation to diastolic blood pressure between SGA and AGA group. Small for gestational age children had significantly higher values of diastolic blood pressure. Systolic blood pressure was similar in both groups. 

An interesting conclusion was reached by a team of researchers from Augusta University, Georgia. The researchers found that adiponectin levels in healthy individuals decrease with increasing obesity and was inversely related to blood pressure and increased with weight loss. [17].In our group of patients none of children were obese but adiponectin serum levels were significantly higher in small for gestational age group compared to appropriate for gestational age group.

Regarding to induced by obesity hypertension the only statistically significant correlation we found between adiponectin levels in SGA group and systolic blood pressure measured oscillometric method. 

Serum resistin levels were found to be higher in European hypertensive patients than in normal pressure controls [50]. Few studies have assessed the relationship between adipocytokines serum profile and blood pressure in the paediatric population.

Our study showed inconsistent results for resistin, which may be due to a small group of patients.

Duncan et al. in their study found no association between serum leptin levels and systolic blood pressure. In contrast, serum adiponectin levels decreased in all groups and were negatively associated with systolic blood pressure [51]. In contrast, it has shown in some work that adiponectin levels are inversely related to blood pressure [17]. We also showed negative correlation between adiponectin levels and systolic blood in our small for gestational age patients. 

In our study we found statistically not significant positive correlation between leptin levels and systolic and diastolic blood pressure in children small as well as appropriate for gestational age.

Also Karakosta et al. in their study did not show an association between cord blood leptin levels and blood pressure [52].

## 5. Conclusions

In summary, children born as SGA have abnormal adipose tissue biomarkers profiles which indicates their dysfunction. The major limitation of our study was a relatively small group of patients. However, this was a single-centre study, which could constitute the beginning of a wider research study to clarify the existing problem.

The results of our study indicate that being born SGA is associated with changes in serum markers of adipose tissue such as adiponectin and leptin. In addition, we have shown a positive correlation between leptin level and current weight as well as a negative correlation between adiponectin level and BMI and current body weight in this group, which may confirm that these markers are involved in the regulation of body weight. As mentioned earlier, SGA children are at greater risk of developing metabolic diseases and their complications, such as hypertension. Small for gestational age children had significantly higher values of diastolic blood pressure and there was significant negative correlation between serum level of adiponectin and systolic blood pressure in oscillometric measurements in this group of patients. Monitoring the profiles of adipokines in SGA patients could be demonstrate early markers for the risk of metabolic diseases in this group of patients. The application of preventive measures could prevent the development of metabolic diseases and their complications. However, assessing the usefulness of adipokine monitoring requires further studies on a larger group of patients.

## Figures and Tables

**Figure 1 nutrients-15-00868-f001:**
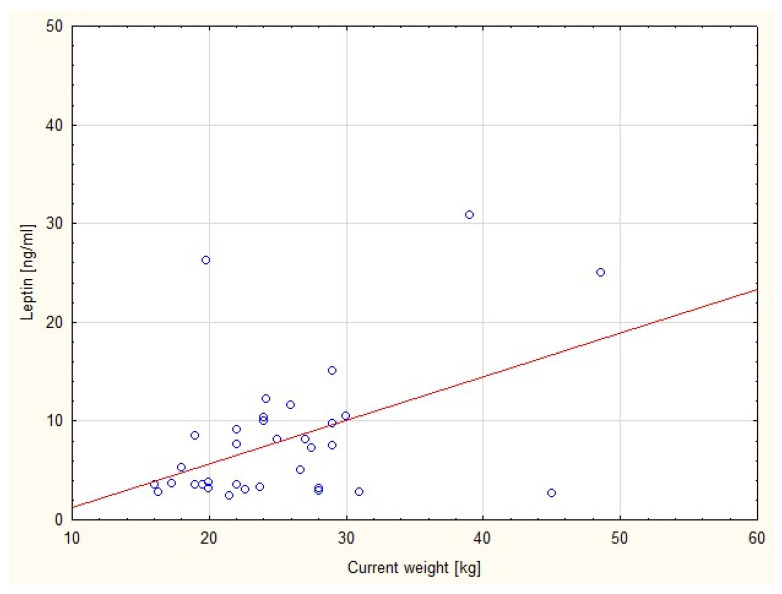
Correlation between serum leptin level and current body weight in SGA group.

**Figure 2 nutrients-15-00868-f002:**
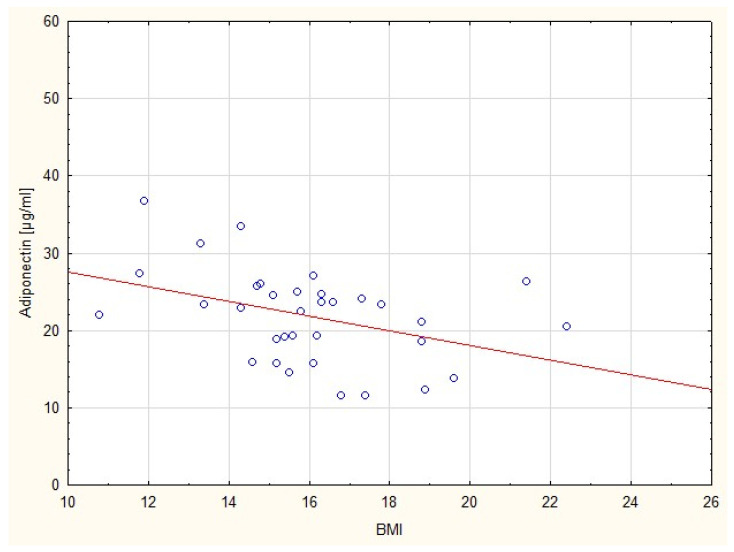
Correlation between serum adiponectin level and BMI in SGA group.

**Table 1 nutrients-15-00868-t001:** Patients’ characteristics.

Parameter	SGA ^a^ Group (N ^c^ = 35) Median (25th Percentile, 75th Percentile)	AGA ^b^ Group (N ^c^ = 25) Median (25th Percentile, 75th Percentile)	*p* Value
**Sex (M ^e^/F ^f^)**	17/18	11/14	NS ^d^
**Age on examination**	8 (7, 9)	9 (8, 12)	NS ^d^
**Birthweight (g)**	2700 (2480, 2730)	3100 (3000, 3550)	***** *p* < 0.001**
**Gestational age (hbd ^g^)**	39 (38, 40)	39 (38, 40)	NS ^d^
**Current weight (kg)**	24 (20, 28)	23.7 (21.2, 23.9)	NS ^d^
**BMI ^h^ (kg/m^2^)**	15.7 (14.7, 17.3)	14.4 (13.7, 15.4)	***** *p* < 0.005**

SGA ^a^—small for gestational age, AGA ^b^—appropriate for gestational age, N ^c^—number of children, NS ^d^—not significant, M ^e^—male, F ^f^—female, hbd ^g^—weeks of gestation, BMI ^h^—body mass index.

**Table 2 nutrients-15-00868-t002:** Adipocytokines levels.

Parameter	SGA ^a^ Group (N ^c^ = 35) Median (25th Percentile, 75th Percentile)	AGA ^b^ Group (N ^c^ = 25) Median (25th Percentile, 75th Percentile)	*p* Value
Adiponectin [µg/mL]	22.84 (18.54, 24.96)	18.85 (10.16, 21.93)	*** *p* < 0.05 (*p* = 0.0134)**
Leptin [ng/mL]	5.33 (3.16, 9.98)	4.20 (1.57, 6.50)	*** *p* < 0.05 (*p* = 0.0415)**
Resistin [ng/mL]	2.07 (1.74, 2.74)	1.94 (1.66, 2.69)	NS ^d^ (*p* = 0.8858)

SGA ^a^—small for gestational age, AGA ^b^—appropriate for gestational age, N ^c^—number of children, NS ^d^—not significant.

## Data Availability

The data used to support the findings of this study are included within the article.

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
