# Peer review of "Serum Adipocytokines Profile in Children Born Small and Appropriate for Gestational Age—A Comparative Study"

_nutrients, 2023, doi:10.3390/nu15040868_

Round 1
Reviewer 1 Report
In this paper, the authors investigate the adipocytokines profile in children born small compared to children appropriate for gestational age.
Although the topic is not novel and the results are expected (a significant number of studies have dealt with this issue), the authors emphasize the importance of their results because there have been no papers on this topic in the last ten years. However, some parts of the work need to be improved and completed with additional information.
The introduction needs to be supplemented with data on the role of adipocytokines in body development, their relationship with adipose tissue and inflammation. Also, it is necessary to state in the introduction the connection between low or high values of adipocytokines and the development of certain metabolic diseases. It would undoubtedly help to understand somehow the risks of the mentioned adipocytokine profile in children born small. Furthermore, the authors failed to mention that children born small have higher risk factors for the development of metabolic syndrome in adulthood. In addition, it is not stated why prepubescent children were chosen, that is, whether the concentration of adipocytokine changes with growth and whether it is affected by sex hormones.
Line 46-50, the goal of the study is written twice!
In the material and method, it is necessary to specify the minimum detectable concentrations or the sensitivity of the ELISA kit for each of the measured adipokines
In the discussion, revising the part related to the results from reference 9 (lines 153-155) is necessary. Namely, the authors discuss the concentration of adiponectin, although the paper cited analyzed the values of resistin, not adiponectin.
In conclusion, the authors should emphasize the importance of the obtained results in light of the risk of developing certain diseases in children born small and perhaps the importance of monitoring these adipokines.
Author Response
Thank you for all your valuable comments and suggestions. Where possible, we have tried to make any corrections or clarify inaccuracies.
Response 1: Thank you very much for your suggestions. Changes have been made to the text
Response 2: Thank you very much for your suggestions. Changes have been made to the text.
Response 3: Thank you very much for your suggestions. Changes have been made to the text.
Response 4: Reference 9 (lines 153-155) relates to resistin concentration, which is discussed in the quoted part of the paragraph (lines 150-155). In lines 150-155, adipokine concentrations are not discussed, but the relationship concerning resistin is covered.
Response 5: As mentioned earlier, children with SGA are at greater risk of developing metabolic diseases and their complications. Monitoring the profile of adipokines in SGA patients could be an early marker for the risk of metabolic diseases in this group of patients. The application of preventive measures could avoid the development of metabolic diseases and their complications. However, assessing the usefulness of the use of adipokine monitoring requires further studies on a larger group of patients.
Reviewer 2 Report
The authors aimed at determining the levels of leptin, adiponectin, and resistin in pre-pubertal children born small or appropriate for gestational age. The 3 adipokines selected are found also in umbilical cord and reported to be associated with birth weight and infant adiposity. The authors found that children born small had higher serum leptin and adiponectin levels compared to the appropriate weight counterparts at the same age. Besides, the serum adiponectin level is only negatively associated with BMI in SGA group , but not AGA group. The manuscript is well written. Major concerns are:
1. Lines 73 and 75--Please specify which systemic diseases and what types of drugs you screened. Does it also include obesity? Also, for the children, do you also screened for the drugs/antibiotics used?
2. Table 2--You specified all p values beyond 0.0001 in other places. Please also specify in this table if p is between 0.0001 and 0.05.
3. Line 108--P=0.108 is not significant
4. Line 112--Data in AGA group should still be reported as a comparison.
5. Figures 1 and 2--Why correlating leptin level with current body weight but adiponectin level with BMI?
6. Is there any difference between sex?
Overall, the sample size is too small, including only a single center (no comparisons among regions or ethnic groups, also no information for genetic background or life style/diet), and this is only a cross-sectional study at a single time point. Besides, with such simple analyses, it is hard to give a convincing conclusion. The authors at least should elaborate more of the data, e.g. display all data in parallel from the AGA group, regardless of the significance, to show the comparisons in values. Since the adipokines are secreted from adipose tissue, though you saw no difference in BMI, is there any difference in waist circumference and body fat/lean mass? What is the indication of the difference in adipokines between AGA and SGA groups? Do you want to use it to explain the reason why they were born small or do you want to use them as biomarkers for later life metabolic abnormalities if there is any?

Author Response
Thank you for all your valuable comments and suggestions. Where possible, we have tried to make any corrections or clarify inaccuracies.
Response 1: We qualified children of healthy mothers before and during pregnancy for the study. Chronic diseases present in mothers that disqualified their children from participation in the study included hypertension, diabetes, rheumatological diseases, thyroid diseases, chronic kidney diseases, obesity and other diseases meeting the criteria for chronic disease. We did not exclude children who had a history of taking antibiotics or other symptomatic drugs used for infections. On the other hand, children with chronic diseases such as diagnosed renal or kidney failure, hypertension, gastrointestinal, nephrological, neurological or cardiac diseases were not included in the study.
Response 2: Thank you very much for your suggestions. Changes have been made to the text.
Response 3: We are very sorry for this mistake. Changes have been made to the text.
Response 4: Thank you very much for your suggestions. Changes have been made to the text.
Response 5: We correlated both adipokines with current body weight and BMI - changes have been made to the text.
Response 6: Due to the small size of the groups, we did not perform an analysis by sex.
Response 7: We are aware of the limitations of our study. Patients who was SGA in the neonatal period were invited to participate in the study. Unfortunately, not all invited patients wanted to participate in the study. Hence the small number of patients eligible for the study. The study centre is located in Lodz, the diversity of ethnic groups within our city is small, which, with regard to the small group of SGA study patients, gives little chance for ethnic comparison of patients. In our study we did not investigate waist circumference and body fat/lean mass. Children with SGA are at greater risk of developing metabolic diseases and their complications. Monitoring the profile of adipokines in SGA patients could be an early marker for the risk of metabolic diseases in this group of patients. The application of preventive measures could avoid the development of metabolic diseases and their complications. However, assessing the usefulness of adipokine monitoring requires further studies on a larger group of patients.
Round 2
Reviewer 1 Report
Dear authors,
thank you for your answers and improvement that you made in the manuscrupt! I will recommend it for publication...